# Health professionals' knowledge on vaccine cold chain management and associated factors in Ethiopia: Systematic review and meta-analysis

**Abebaw Wasie Kasahun**[1]\*, **Amare Zewdie**[1]\*, **Ayenew Mose**[2], **Haimanot Abebe Adane**[3]

**1** Department of Public Health, College of Medicine and Health Science, Wolkite University, Wolkite, Ethiopia, **2** Department of Midwifery, College of Medicine and Health Science, Wolkite University, Wolkite, Ethiopia, **3** Healthy working Lives Research Group, School of Public Health and Preventive Medicine, Monash University, Melbourne, Victoria, Australia

\* abebawasie@gmail.com (AWK); amarezewdie23@gmail.com (AZ)

**Data Availability Statement:** All relevant data are within the paper and its Supporting Information files.

## Abstract

### Background

Vaccines are playing a great role in prevention of many infectious diseases worldwide. Health professionals' knowledge towards vaccine cold chain management is an essential element of maintaining vaccine's potency at shelf and during transportation. Studies on health professionals' knowledge on vaccine cold chain management system and associated factors in Ethiopia have inconclusive findings. This systematic review and meta-analysis is aimed to produce the overall level of health professionals' knowledge on vaccine cold chain management system and to identify its associated factors in Ethiopia.

### Methods

Systematic review and meta-analysis was conducted on health professionals' knowledge on vaccine cold chain management in Ethiopia. It is registered under PROSPERO website with registration number CRD42023391627. Literature search was made on international data bases using medical subject heading and key words. Data were extracted using Microsoft excel and imported to STATA version 17 for analysis. Heterogeneity was checked using Cochrane Q test and $I^2$ statistics. Weighted Inverse variance random effect model was used to estimate the pooled level of health professionals' knowledge on vaccine cold chain management. Publication bias was checked using funnel plot and using Egger's test.

### Results

A total of nine studies were included in the review. The pooled health professionals' good knowledge on vaccine cold chain management in Ethiopia is 49.92% with 95% CI (48.06–51.79). Having five years or more experience AOR 2.27 95% CI (1.72–2.99), being nurse AOR 3.03 95% CI (1.47–6.27), received on job training AOR 6.64 95% CI (4.60–9.57), EPI guideline available at facility AOR 2.46 95% CI (1.75.-3.48) are factors positively associated with health professionals' knowledge on vaccine cold chain management in Ethiopia.

**Funding:** The authors received no specific funding for this work.

**Competing interests:** The authors have declared that no competing interests exist.

**Abbreviations:** EPI, Expanded program on Immunization; Fig, Figure; PICO, Population, Intervention, Comparison, outcome; PRISMA, Preferred Reporting Items for Systematic Reviews and Meta-Analyses; MeSH, Medical subject heading; SNNPR, South Nations, Nationalities and People Region; VPDs, Vaccine Preventable Diseases; WHO, World Health Organization.

## Conclusion

The pooled prevalence of good knowledge on vaccine cold chain management among health professionals is much lower than the expected level. There is a need to plan on job trainings for all vaccine handlers and other health professionals supposed to work on vaccination program.

## Introduction

The world has made remarkable progress in child survival in the past three decades, and millions of children have better survival chances now than in 1990 (1 in 27 children died before reaching age five in 2020, compared to 1 in 11children in 1990). Though child mortality is declining worldwide, there is still high level of preventable under-five mortality. Globally, ten million children are dying annually before celebrating their fifth birth day; of which the majority deaths happened in developing regions [1,2].

A number of public health interventions have been implemented for improving child survival worldwide. Immunization is one of the most effective strategies for preventing vaccine preventable infectious disease and associated child deaths [3]. According to the World Bank report, immunization program is saving over two million children's life annually in the world. In spite of this fact, still vaccine preventable diseases attribute 25% of under-five mortality worldwide [4,5]. Despite the introduction of the lifesaving vaccines, vaccine-preventable diseases (VPDs) still accounts for the death of more than half a million children under five year olds every year in Africa representing 58% of global VPD-related deaths [6].

Vaccine cold chain management system is one of the components of vaccination program to maintain potency of vaccines which in turn will reduce burden of vaccine preventable diseases. Administration of potent vaccine in a manner of well-maintained cold chain system and achieving high vaccine coverage is one of the public health focus area in developing regions of the world. However, Vaccine cold chain management system is the most challenging factors for the effectiveness of expanded program of immunization [7].

Vaccines are thermo-sensitive and have a fixed shelf time with a possible loss of viability over time. This loss is irreversible and accelerated if proper storage and temperature conditions are not maintained. The aforementioned biochemical property of vaccines necessitates keeping the potency of the vaccines through a well-functioning cold chain system that satisfies specific temperature requirements of each vaccine [8]. According to the World Health Organization recommendations, vaccines should be stored in a refrigerator keeping the temperature reading between +2 and 8 degree Celsius. Furthermore, vaccine should be held by a designated vaccine carrier with properly prepared coolant packs during transportation. Essential logistics and apparatuses including vaccine refrigerator, electronic refrigerator temperature loggers, vaccine vial monitoring and knowledgeable and skilled vaccine handlers are required for ensuring a well-functioning cold chain system at all-time points [9].

Vaccine cold chain is a system for storing and transporting vaccines within an acceptable temperature range from the manufacturer to users. Cold chain system consists of a series of storage and transport links, all designed to keep vaccines within an acceptable temperature range until it reaches the users. Improper storing and handling of vaccines can let vaccines lost potency, failure to ignite immune responses, will make vaccines more reactogenic and unable to prevent vaccine preventable diseases [10].

Vaccines need more complex handling and storage requirements due to increased thermal sensitivity and complicated immunization schedules. This necessitates adequate training and

supervision to equip vaccine handlers with basic vaccine cold chain maintenance protocols. The vaccine cold chain guideline recommends the following: the vaccine storage should be maintained in the temperature range of 2–8˚C, the use of minimum/maximum thermometers, temperature charts, and the shake test [11,12]. Availability of cold chain equipment, knowledge of cold chain handlers, EPI guideline utilization, education status, in service training and supervision are some of the factors affecting the cold chain management practices [13,14]. Cold chain maintenance system and temperature monitoring is still a major concern in developing countries where both coverage and quality of immunization service is suboptimal [15].

In Ethiopia, vaccines transported and stored at different points before reaching users. The vaccine cold chain starts at ministry of health and go through regional health bureau, zonal health desks, district health offices, health centers and health posts. Throughout the chain, health care providers must have adequate knowledge on how to manage the cold chain [12]. If the cold chain is broken at any points of the aforementioned stakeholders, vaccine will lose their potency. Dysfunctional cold chain system will inhibit all other efforts tailored to reduce child mortality and morbidity. Thus, health facilities and health offices that handle vaccines have to be equipped with apparatuses such as refrigerator specifically designed for storing vaccines, in addition it is essential to equip health professionals with necessary knowledge and skills of vaccine cold chain management system [16].

Effective maintenance of cold chain system at all levels is essential to keep vaccines potent and used to narrow the gap between vaccinated and immunized children. It is also crucial for reducing vaccine wastages. Good knowledge of health professionals towards effective vaccine cold chain management is one of the essential attributes to keep vaccines potent. It is essential to determine the level of knowledge, attitudes and practices of health care providers working at the grass root level of vaccination and the cold chain system to take corrective measures at an early stage, however, the level of health professionals' knowledge on vaccine cold chain management system in Ethiopia is not unequivocally determined. Previous researches have reported divergent levels of health professionals' knowledge across different levels of the health sectors. Besides, the determinants for the observed differences in the health professionals' level of knowledge on vaccine cold chain management are also different across different settings [17–19].

Ethiopia has gained a steady increment in coverage of all basic vaccinations. According to the national health and demographic surveys, coverage of all basic vaccinations increased from 39% in 2016 to 43% in 2019 [20,21]. Though significant progress has been made in increasing vaccine coverage in Ethiopia, Vaccine preventable diseases outbreaks are still being reported in different parts of the country [22,23]. One of the speculations made for the occurrence of vaccine preventable diseases is poor vaccine cold chain management system at different levels of the health sector from vaccine procurement, storage, handling and transporting to users that might lead vaccines lose potency and failure to produce immune responses for preventing infections.

In Ethiopia, there is no unequivocal level of health professionals' knowledge on vaccine cold chain management system. Previous studies on health professionals' knowledge on vaccine cold chain management system were conducted in a fragmented way across different subnational settings that cannot represent the diverse national contexts. As a result, the level of reported health professionals' knowledge on vaccine cold chain management have shown divergent findings and associated factors are also inconsistent across different studies [17–19,24,25]. Therefore, the aim of this systematic review and meta-analysis is to produce the pooled level of health professionals' knowledge on vaccine cold chain management system and to identify the most common factors associated with health professionals' knowledge on vaccine cold chain management in Ethiopia.

## Methods

### Study design and setting

A systematic review and meta-analysis was conducted to determine the pooled level of health professionals' knowledge on vaccine cold chain management and to identify its associated factors in Ethiopia. Initially, the PROSPERO website (http://www.library.ucsf.edu/) has been checked whether the title is previously addressed or an ongoing review exists to avoid duplications. Accordingly, we found no registered published and/or ongoing systematic review and meta-analysis on determinants of health professionals' knowledge on vaccine cold chain in Ethiopia. The protocol of this systematic review and meta-analysis is registered in PROSPERO data base with registration number CRD42023391627. The review was conducted in accordance with the preferred reporting items for systematic review and Meta–analysis (PRISMA) checklist [26].

### Search strategies and source information

First preliminary search was done using medical subject headings. Secondly, key words were developed using key words from retrieved articles on the preliminary search. Finally, medical subject headings and key words were used to search articles on medical and health sciences research data bases and other search engines. Furthermore, librarians were consulted to find unpublished research works on our area of interest for this review. Data bases including PubMed, Scopus, African Journals Online, Web of Science, Google scholar and Google were used to find research articles on health professionals' knowledge towards vaccine cold chain management in Ethiopia. Furthermore, hand search was made on references of retrieved articles to find all eligible articles for this review.

Search terms were designed using CoCoPop/PEO guideline. Searching of articles was done using medical subject headings (MeSH) and key terms through online data bases. Boolean operators including "AND" and "OR" were used to link MeSH and the key words for searching purpose. Accordingly, the developed search terms consist of "Vaccine Cold chain" OR "cold chain" AND "knowledge" AND "determinants" OR "associated factors" OR "predictors" AND "Ethiopia".

### Eligibility criteria

Studies reporting the prevalence of good knowledge of vaccine cold chain management and associated factors among health professionals were included in this review. Published research articles and unpublished researches including preprints and gray literatures written in English language were eligible regardless of the study design. Articles published at any time until the end of our search (January 20, 2023) were included in this review. Articles which did not report the outcome variable or articles with unrelated outcome variable to the interest of this review were excluded. Furthermore, articles without full abstracts, commentaries, editorials, letters and anonymous reports were excluded.

### Outcome measurement

The level of health professionals' knowledge on vaccine Cold chain management and factors associated with health professionals' knowledge on vaccine cold chain management are the two outcome variables of this review. Cold chain is a system of people and equipment which ensures potent vaccine reach the users. Cold chain is a network of refrigerators, cold boxes and vaccine carriers for keeping vaccines within the recommended temperature range ($2\,^{0}$C- $8\,^{0}$C) to safeguard their potency during transportation, storage and delivery. Cold chain is said to be

properly maintained if and only if there is proper temperature monitoring (twice daily monitoring), proper review of vaccine vial monitoring to check exposure of vaccine to heat, undertaking shake test to check exposure of vaccine to freezing temperature, adherence to the first expiry first out vaccine flow in distributing vaccine to users, regular cleaning and defrosting of refrigerator/fridge ice and timely cold chain equipment maintenance during breakage as per WHO recommendation. The knowledge of health professionals regarding the above mentioned vaccine cold chain maintenance protocol was assessed by asking them different items. Total knowledge score was obtained after adding each response (if they correctly respond for the item, they got 1 or else 0 point). The mean score of respondents on knowledge assessment items were computed, and health professionals who scored below the mean score were labeled as having poor knowledge, while those who scored the mean and above the mean value were labeled as having a good knowledge on vaccine cold chain management.

## Data extraction

Data were extracted using Joanna Briggs institute data extraction form for observational studies [27]. Firstly, identified articles were imported to EndNote X6 for the purpose of identifying and removing duplicated articles. Secondly, important data were extracted using the prepared data extraction format independently by two authors (AZ and AW). For the first outcome of the review the extracted data includes primary author name, study year, publication status, publication month, publication year, study design, sample size, prevalence, study region, determinant factors and quality of the study. Data were extracted using 2 by 2 tables for the second objective of this review (associated factors of health professionals' knowledge on cold chain management). All data extraction activity was done by two authors (AW and AZ). Finally, data analysis was done by STATA software version 17.

## Quality assessment

Modified Newcastle Ottawa quality assessment scale for cross-sectional studies was used to assess quality of studies [28]. Two authors (AW and AZ) assessed the quality of each study using the aforementioned quality assessment scale. The quality assessment scale includes methodological quality, sample selection, sample size, outcome measurement and statistical analysis used. In cases of disagreement between the two authors the third author (GA) involved for resolution.

## Synthesis methods and reporting bias assessment

Data were extracted using the Microsoft excel spreadsheet format and imported to STATA software version 17 for analysis. Heterogeneity among studies was checked using Cochrane Q test and $I^2$ statistics. The level of heterogeneity among studies is quantified by $I^2$ statistics. Accordingly, if the result of $I^2$ is 0% to 40% it is mild heterogeneity, 30 to 60% would be moderate heterogeneity, 50 to 90% would be substantial heterogeneity; and 75 to 100% would be considerable heterogeneity [29]. Weighted Inverse variance random effect model was used to estimate the pooled level of health professionals' knowledge on vaccine cold chain management system in Ethiopia. Random effect model was used due to the observed substantial heterogeneity ($I^2$ = 97.3%) among studies. Forest plot was used to illustrate the pooled level of health professionals' knowledge on vaccine cold chain with 95% CI. Publication bias was checked visually using funnel plot and statistically using Egger's regression test, with P<0.05 indicating significant publication bias. Sensitivity analysis was done to estimate the effect of single study on the overall estimate of the level of health professionals' knowledge on vaccine

cold chain management and sub-group analysis was done using regions where the studies are conducted.

## Certainty assessment

Grading of Recommendations Assessment, Development and Evaluation (GRADE) assessment was used to assess the overall certainty of the evidence. A GRADE assessment comprises risk of bias to the internal validity of results, consistency of results across studies, directness and precision of results, and likelihood of publication bias. The overall quality of evidence is then categorized as high, moderate, low or very low [30]. Grading of Recommendations Assessment, Development and Evaluation assessments were conducted for the primary outcome included in the meta-analysis. Two independent researchers (AWK and AZ) performed the GRADE assessments.

## Results

The search strategy yielded a total of 164 articles about health professionals' knowledge on vaccine cold chain management in Ethiopia. Duplicates (79 articles) identified by endnote software and removed. From the remaining 85 articles 53 articles were excluded by reading the titles. The remaining 32 articles sought for full article retrieval and 30 full articles retrieved and 2 articles excluded because of unable to retrieve the full article. Thirty articles were assessed for eligibility and 21 articles were excluded due to reasons explained in Fig 1. Accordingly, nine articles have fulfilled the inclusion criteria and included in the final systematic review and meta-analysis. From the included nine studies three were from Amhara region of Ethiopia, three from SNNPR of Ethiopia, two from Oromia region of Ethiopia and one nationwide. All of the included studies have used cross-sectional study design and included sample size ranges from 116 to 632 health professionals (Table 1).

### Health professionals' knowledge on vaccine cold chain management in Ethiopia

The pooled level of health professionals' knowledge on vaccine cold chain management in Ethiopia is 49.92% with 95% CI (48.06–51.79). Substantial level of heterogeneity is observed with $I^2$ statistics ($I^2$ = 97.3%, p = 0.000). The finding is summarized using forest plot presented below in Fig 2. Based on the findings of substantial heterogeneity sub group analysis was done using regions where the studies are conducted.

### Publication bias

Funnel plot and egger's regression test was used to assess whether publication bias is observed or not, accordingly, the egger's test was not statistically significant (p = 0.36) that rule out the presence of publication bias. Furthermore, funnel plot is symmetrical which confirms the absence of publication bias (Fig 3).

**Sub-group analysis of health professionals' knowledge on vaccine cold chain.** The subgroup analysis showed that the overall level of health professional knowledge is highest nationwide with 61.55% 95% CI (57.76–65.34) and lowest in Amhara region with pooled level of good knowledge of health professionals on vaccine cold chain management 31.27 95% CI (27.87–34.67) (Fig 4).

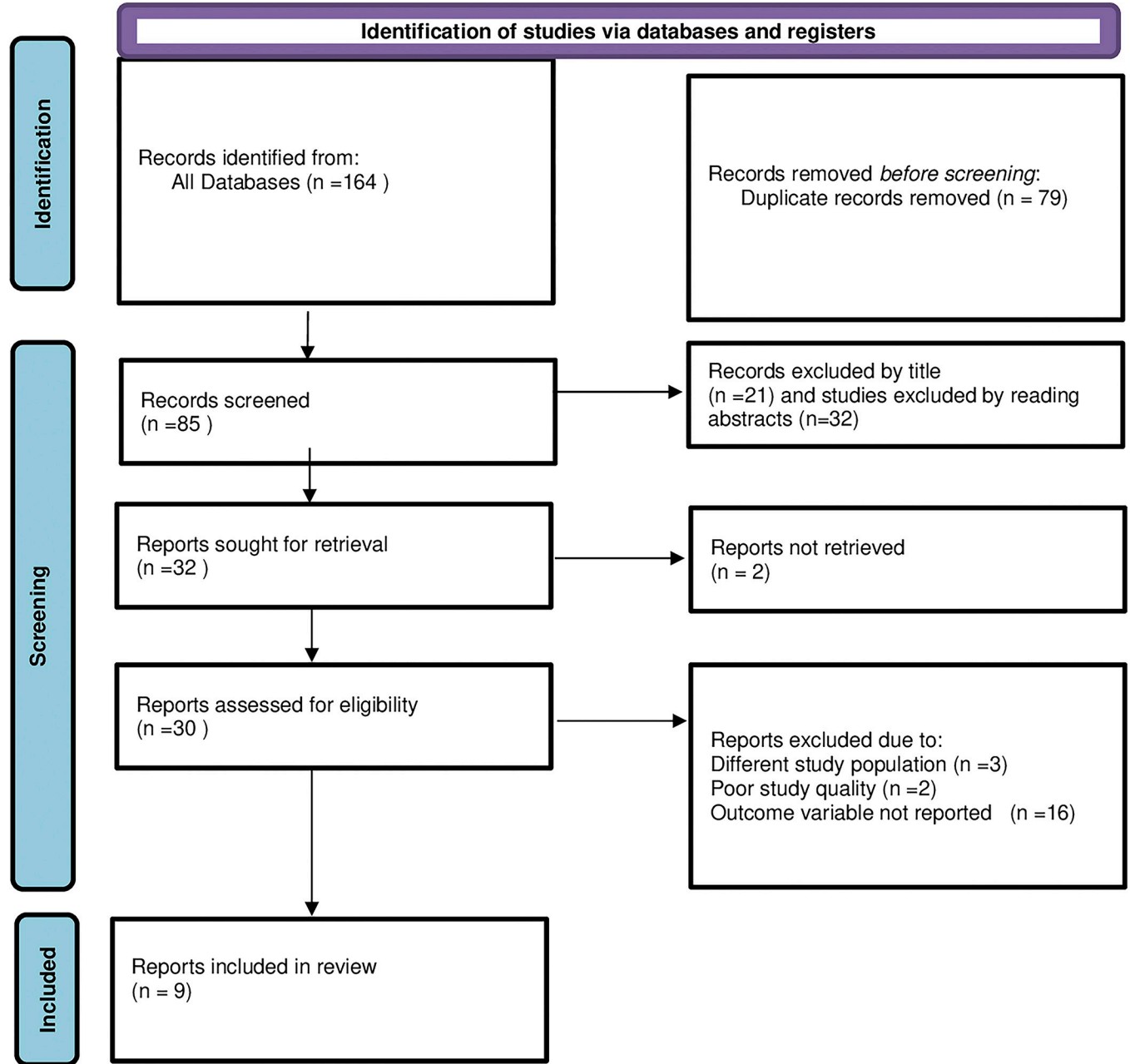

**Fig 1. Flow chart illustrating selection of studies for systematic review and meta-analysis on knowledge of health professionals on vaccine cold chain management in Ethiopia, 2023.**

## Sensitivity analysis

A random effect model finding illustrated that no single study has influenced the pooled level of health professionals' knowledge in Ethiopia (Fig 5).

**Factors associated with health professional's knowledge on vaccine cold chain management in Ethiopia.**    Health professionals' years of experience, profession category, and experience of on job training on vaccine management and availability of EPI guideline at the health facility were significantly associated with health professionals' knowledge on vaccine cold chain management.

**Table 1. Characteristics of included studies for systematic review and meta-analysis on knowledge of health professionals' on vaccine cold chain management in Ethiopia.**

| Author | Study year | Study region | Study design | Sample size | Prevalence of good knowledge | Study quality |
|---|---|---|---|---|---|---|
| Asres M et al [18] | 2019 | Nationwide | Cross-sectional | 632 | 61.5% | good |
| Degavi G et al [31] | 2021 | Oromia | Cross-sectional | 502 | 50.4% | good |
| Feyisa D et al [32] | 2022 | SNNPR | Cross-sectional | 140 | 53.5% | good |
| Mohammed SA et al [33] | 2021 | Amhara | Cross-sectional | 127 | 53.5% | good |
| Rogie B et al (unpublished) [34] | 2012 | Amhara | Cross-sectional | 116 | 56% | good |
| Woldemichael B et al [35] | 2018 | Oromia | Cross-sectional | 183 | 54.6% | good |
| Yassin ZJ et al [36] | 2019 | SNNPR | Cross-sectional | 232 | 51.3% | good |
| Erassa TE et al [37] | 2023 | SNNPR | Cross-sectional | 136 | 78.7% | good |
| Zelalem E et al(unpublished) [38] | 2020 | Amhara | Cross-sectional | 396 | 21.7% | good |

Health Professionals with more than five years of experience are 2.27 times more likely to have good knowledge on vaccine cold chain management 2.27 95% CI (1.72–2.99). Nurse health professionals are 3.03 more likely to have good knowledge towards vaccine cold chain management with 95% CI 3.03(1.47–6.27).

Health professionals who have on job training are 6.64 times more likely to have good knowledge compared to those who didn't get on job training with 95% CI 6.64 (4.60–9.57). Health professionals working in health facilities where EPI guideline is available are 2.46 times more likely to have good knowledge towards vaccine cold chain management compared to those who works in health facilities where EPI guidelines is not available with 95% CI 2.46 (1.75.-3.48) (Table 2).

**Assessment of certainty.** As all of the included studies are cross sectional, we stood at low certainty of evidence and further assessed certainty the evidence using the five down grading and three up grading elements of the GRADE certainty assessment criteria [30]. Accordingly, the generated evidences tend to suffer from inconsistencies due to the fact that few studies are included in the review. It is demonstrated by significantly identified heterogeneity of estimate the outcome variable across different administrative regions of Ethiopia. However, there is no identified effect of publication bias and risk of bias as all included studies have followed robust methods and egger's test and funnel plot demonstrated the absence of publication bias. There

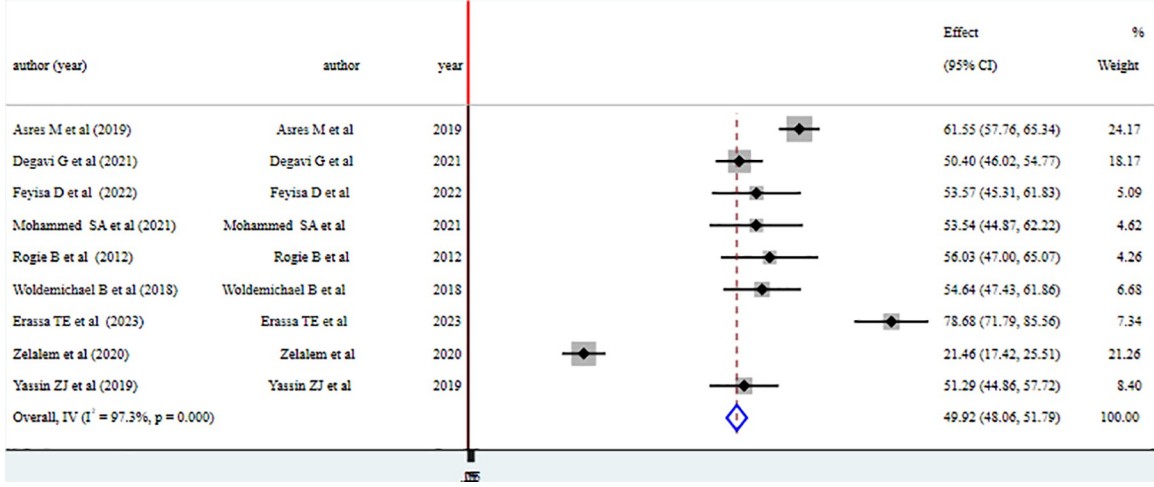

**Fig 2. Pooled level of good knowledge on vaccine cold chain management among health professionals in Ethiopia, 2023.**

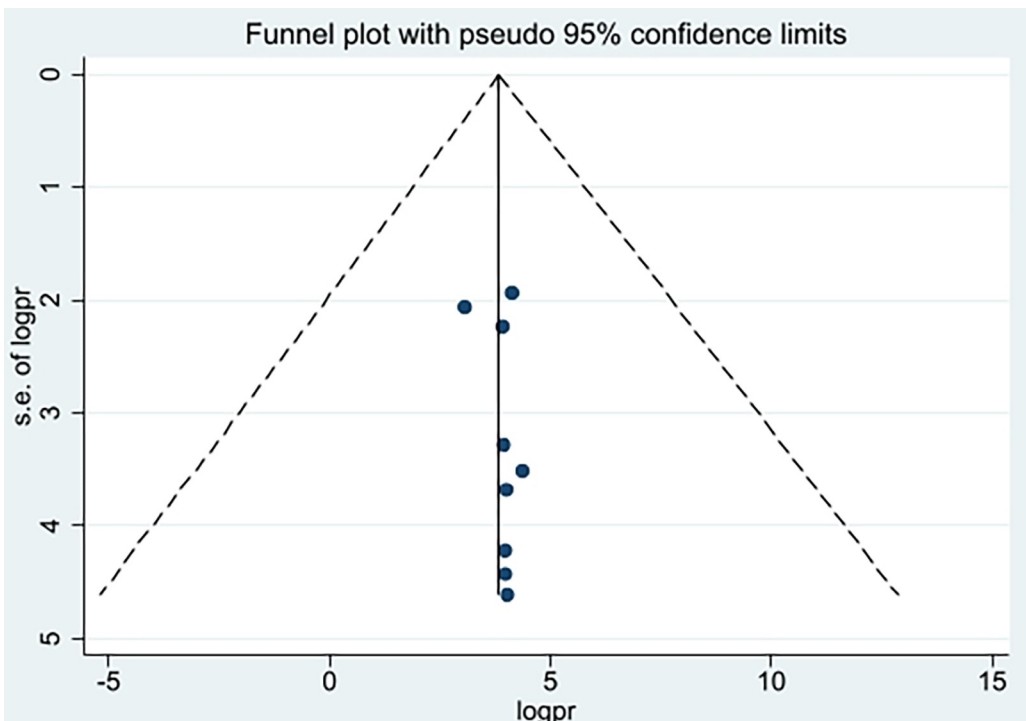

**Fig 3. Funnel plot showing the symmetric distribution of articles on health professionals' knowledge on vaccine cold chain management in Ethiopia, 2023.**

is no evidence of imprecision of the measurements of the outcome variable as the sample size is well enough for all included studies and the confidence interval of the estimated pooled prevalence is narrow. On top of that, the outcome variable is directly measured from all the included studies, no included study has indirectly measured the outcome variable of this review. The confounding variables are controlled by considering all the factors that may affect measurements of the estimates of variable and the estimate of the outcome variable is large enough in all included studies.

In general, the current estimate of health professionals' knowledge on vaccine cold chain management in Ethiopia is highly likely to change if more studies are included; hence it warrants further studies to produce more accurate evidence that we can rely on for decision making purpose.

## Discussion

This systematic review and meta-analysis is aimed to determine the pooled prevalence of health professionals' knowledge on vaccine cold management and associated factors in Ethiopia. Accordingly, only half of health professionals (49.92% with 95% CI (48.06–51.79)) have good knowledge on vaccine cold chain management in Ethiopia. The level of good knowledge on vaccine cold chain management among health professionals has shown considerable variation across different sub-national settings. Health professionals in the Amhara region have the lowest level of good knowledge whereas the study conducted nationwide yielded the highest level of good knowledge on vaccine cold chain management. The reasons for the observed disparity might partly be due to the fact that health institutions' infrastructure such as availability of guidelines, teaching aids and opportunities of on job trainings are relatively good at central

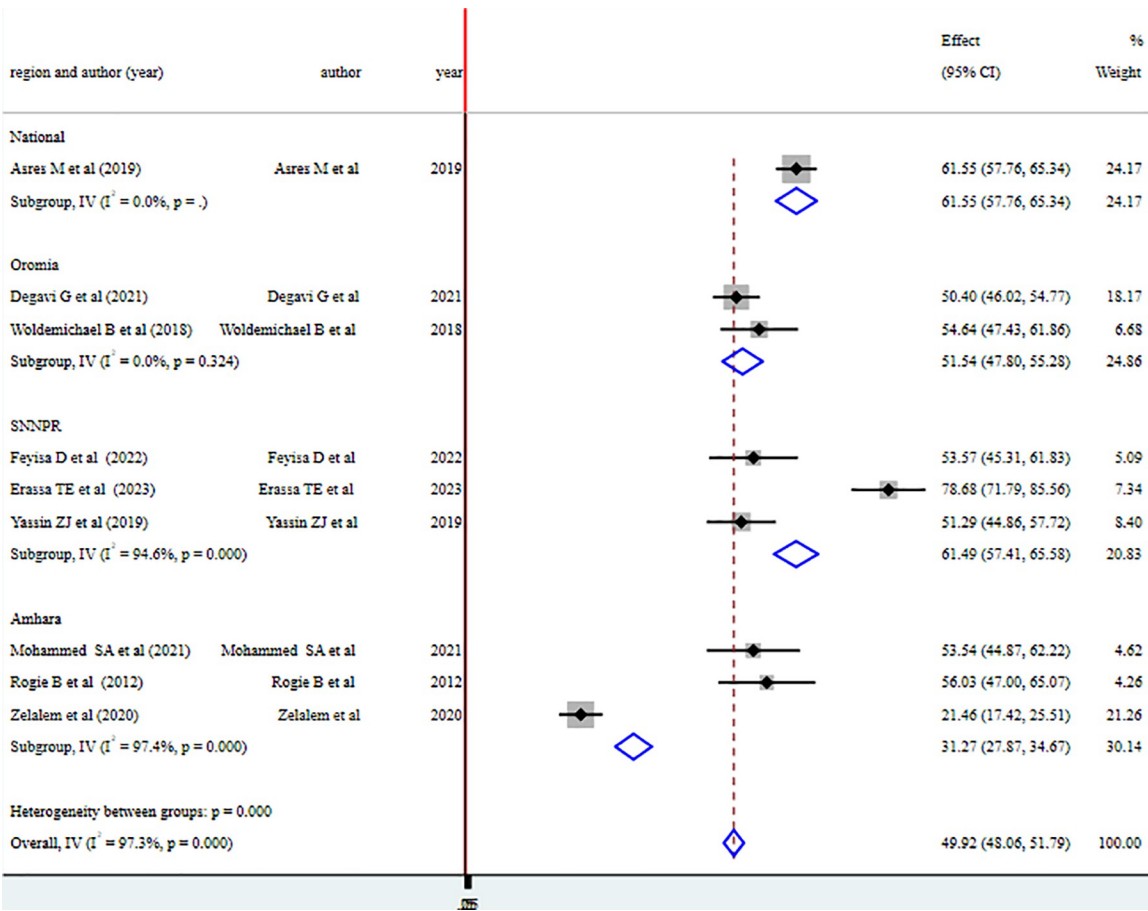

**Fig 4. Forest plot illustrating Subgroup analysis of health professionals' knowledge on vaccine cold chain management by region in Ethiopia, 2023.**

parts of Ethiopia where most included studies have been conducted compared to the Amhara region (Northern part of Ethiopia). The overall prevalence of good knowledge on vaccine cold chain management in this study is higher than the reported level of health professionals' knowledge on vaccine cold chain management in Edo state of Nigeria, Giwa state of Nigeria and India where only 36%, 3.6% and 28.34% of health professionals have good knowledge pertaining to vaccine cold chain management respectively [39–41]. However, it has to be kept in mind that the findings from Nigeria and India were taken from single studies in each country unlike the finding of this study which is pooled prevalence combined from many studies in Ethiopia. The other reason for the observed difference might be due to the small sample size in case of the study in Pradesh, India.

The pooled level of health professionals' knowledge on vaccine cold chain management in Ethiopia is also higher than findings reported in Northwestern region of Cameroon [42]. The difference might be due to the fact that more stringent criteria were used to measure knowledge in the case of a study in Cameroon whereas mean and/or above score for knowledge related items was used in case of studies included in this systematic review and meta-analysis. The later approach will underestimate the level of knowledge and the earlier approach may lead to slight overestimation of health professionals' knowledge on vaccine cold chain management.

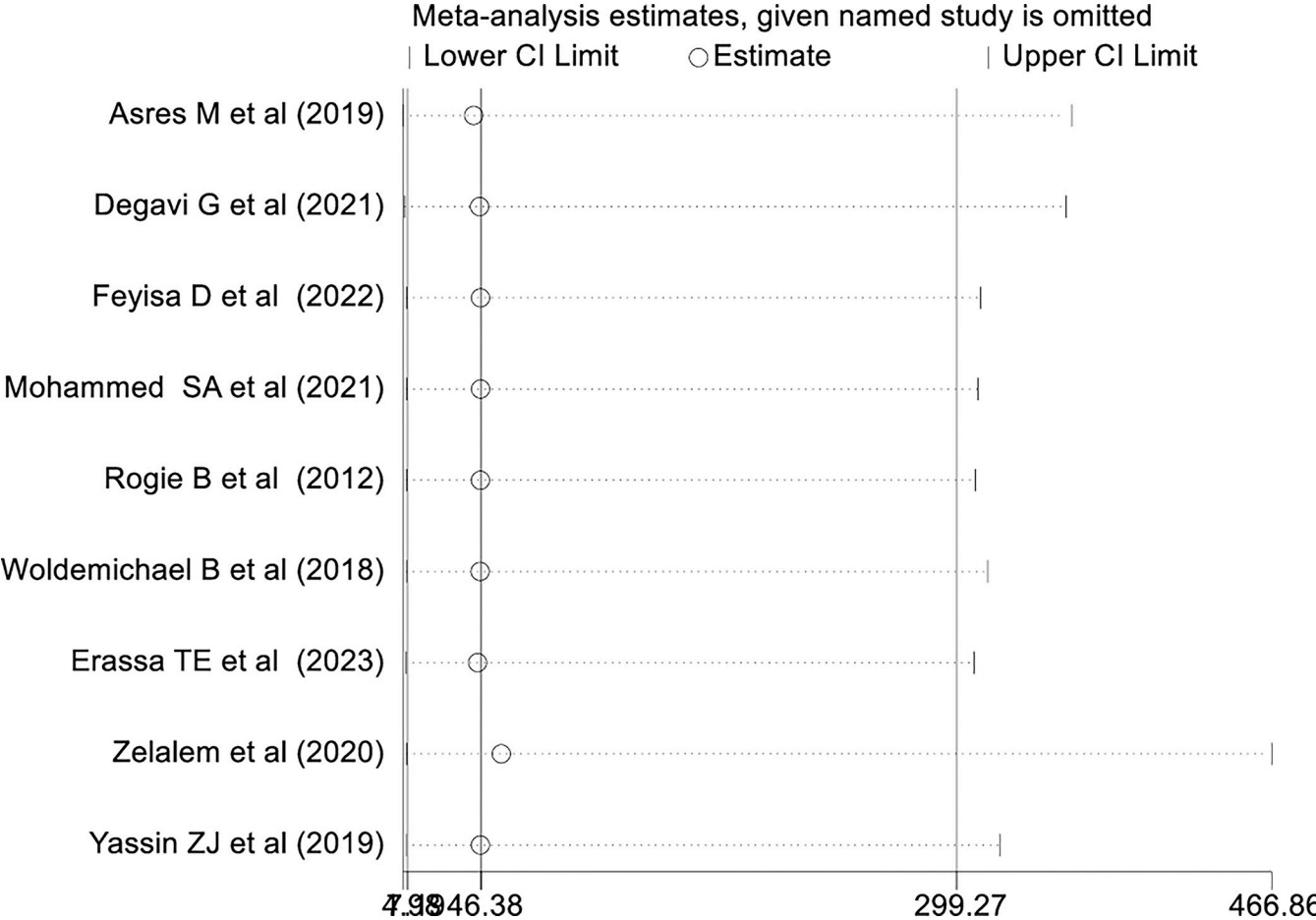

**Fig 5. Sensitivity analysis of studies included for systematic review and Meta-analysis on health professionals' knowledge on vaccine cold chain management in Ethiopia, 2023.**

**Table 2. Factors associated with health professionals' knowledge on vaccine cold chain management in Ethiopia.**

| Variable | Authors | AOR | 95%CI | Pooled OR | 95% CI of pooled OR |
|---|---|---|---|---|---|
| greater than five years of experience | Degavi G et al [31] | 2.2 | 1.99–5.16 | 2.27 | 1.72–2.99 |
| | Yassin ZJ et al [36] | 2.1 | 1.8–4.15 | | |
| | Asres M et al [18] | 2.78 | 1.54–5.01 | | |
| Being nurse health professional | Degavi G et al | 2.12 | 1.46–15.44 | 3.03 | 1.47–6.27 |
| | Rogie B et al [34] | 8.83 | 1.86–41.9 | | |
| | Yassin ZJ et al | 2.4 | 1.47–14.4 | | |
| Received on job training | Degavi G et al | 6.13 | 2.78–11.2 | 6.64 | 4.60–9.57 |
| | Yassin ZJ et al | 5.19 | 2.68–10.11 | | |
| | Mohammed SA et al [33] | 3.04 | 1.04–8.88 | | |
| | Zelalem E et al [38] | 12.28 | 6.33–23.82 | | |
| Working in facility where EPI guideline is available | Degavi G et al | 2.58 | 1.47–4.57 | 2.46 | 1.75–3.48 |
| | Zelalem E et al | 2.17 | 1.11–4.22 | | |
| | Yassin ZJ et al | 2.58 | 1.47–4.57 | | |

On the other spectrum, a study in Osun state of Nigeria, Yemen and Ghana have reported better level of health professionals' knowledge on vaccine cold chain management than the finding of this study [43–45]. The observed difference could be due to the difference in study population, for instance vaccine handlers were the study population for the study in Ghana whereas studies included in this systematic review and meta -analysis were used all health professionals as a study population regardless of their unit of duty at the health facility as a result vaccine handlers' might have more refreshed knowledge on vaccine cold chain than health professionals' working in other units.

Health professionals' years of experience, training on vaccine management, availability of EPI guideline at health facility and type of health profession were significant predictors of health professionals' knowledge on vaccine cold chain management in Ethiopia.

Health professionals who have five years or above service experience in health care are twice more likely to have good knowledge on vaccine cold chain management compared to health professionals with lower years of experience. This is obvious due to the fact that experience helps to refine one's level of knowledge and practices in an individual's ladder of career. It also paves the chance to get trainings on guidelines, standard operating procedures and to accumulate wisdom from many other sources including from colleagues which might help to boost the level of health professionals' knowledge. This finding is consistent with a study in Malaysia where health professionals' service experience is found to be significant predictors of health professionals' knowledge on vaccine cold chain management [46].

Health professionals who have got on job training on expanded program of immunization are about seven times more likely to have good knowledge on vaccine cold chain management compared to professions who didn't get on job training about vaccination. It is known that on job training is helpful to acquire the current state of art in health care including vaccine handling; hence health professionals who have received on job training are more likely to recognize evidence based vaccine handling practices and will help them to refresh and sustain their knowledge on how to maintain potency of vaccines. This finding is consistent with studies conducted in Turkey, Yemen and Malaysia which proves the effectiveness of on job training of health professionals in improving their knowledge and practices of vaccine cold chain management [43,46,47].

Nurse health professionals are three times more likely to have good knowledge on vaccine cold chain management compared to others. This might be due to disproportionate assignment of nurses to units dedicated for EPI and other vaccination programs in Ethiopia. Since there is shortage of general practitioners to be deployed in every unit of public health facilities, EPI units are mostly staffed by nurse health professionals in Ethiopia; as a result they have a chance to get updated manuals, trainings regarding vaccine handling and lessons from their daily activity enables them to be more knowledgeable on vaccine cold chain management. Similar finding is reported in Yemen that shows Nurse Health professionals have better knowledge on vaccine cold chain management than other health profession categories [43].

Health professionals' who works in health facilities where EPI guideline or vaccine cold chain management guideline is available are nearly three times more likely to have good knowledge on vaccine cold chain management compared to those working in facilities where the guideline is not available. Availing guidelines and other related written materials at health facility is crucial to improve health professionals' knowledge as it creates a chance to refer these materials in time of need.

This systematic review and Meta -analysis is not without limitations. The followings are some of the shortcomings that have to be considered in interpreting and using finding of this study. The first drawback is that, the result of this systematic review and meta-analysis is based on few studies; as a result it may not show the true magnitude of the outcome variable. It may

possibly be different if many more studies were done and included in this systematic review and meta-analysis. The second and the final limitation is stemmed from the included studies; the result of this review and meta-analysis solely depends on the already available data from previous studies, consequently, the limitation and quality of the previous researches have an impact on the findings of this study.

## Conclusions

The pooled prevalence of good knowledge on vaccine cold chain management among health professionals is much lower than the expected level in Ethiopia. It is unlikely to maintain the potency of vaccines until it reaches to beneficiaries with the existing level of health professionals' knowledge on protocols of maintaining vaccine potency. Years of experience, availability of guidelines at health facility, being nurse and getting on job training on cold chain management were significant predictors of good health professionals' knowledge on vaccine cold chain management. The finding calls for an urgent intervention to improve health professionals' knowledge on vaccine cold chain management. There is a need to plan on job training on vaccine cold chain management for all vaccine handlers and other health professionals supposed to work on vaccination program. Furthermore, it is also important to avail cold chain management guidelines at each health facility so that health professionals can refer when they need it.

## Supporting information

**S1 Checklist. PRISMA checklist result.**
(DOCX)

**S1 File. Quality assessment result of included studies.**
(DOCX)

**S2 File. GRADE certainty of evidence: Assessment result of certainty of evidence.**
(DOCX)

**S1 Dataset. Minimal data set: Data extracted from included studies.**
(XLSX)

## Acknowledgments

We would like to thank all authors of the primary studies which are included in this systematic review and meta-analysis.

## Author Contributions

**Conceptualization:** Abebaw Wasie Kasahun.

**Data curation:** Abebaw Wasie Kasahun, Ayenew Mose.

**Formal analysis:** Abebaw Wasie Kasahun, Amare Zewdie, Ayenew Mose.

**Writing – original draft:** Abebaw Wasie Kasahun, Amare Zewdie, Ayenew Mose.

**Writing – review & editing:** Amare Zewdie, Haimanot Abebe Adane.

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
