## [Decision Letter · Decision Letter 0]

14 Jun 2023

PONE-D-23-03134Health professionals’ Knowledge on Vaccine cold chain management and associated factors in Ethiopia: systematic review and meta-analysisPLOS ONE

Dear Dr. Kasahun,

Thank you for submitting your manuscript to PLOS ONE. After careful consideration, we feel that it has merit but does not fully meet PLOS ONE’s publication criteria as it currently stands. Therefore, we invite you to submit a revised version of the manuscript that addresses the points raised during the review process.

We look forward to receiving your revised manuscript.

Kind regards,

Demeke Mesfin Belay, MSc

Academic Editor

PLOS ONE

Journal Requirements:

3. Please upload a new copy of Figure 2 and 4 as the detail is not clear. Please follow the link for more information: " ext-link-type="uri" xlink:type="simple">https://blogs.plos.org/plos/2019/06/looking-good-tips-for-creating-your-plos-figures-graphics/"
https://blogs.plos.org/plos/2019/06/looking-good-tips-for-creating-your-plos-figures-graphics

4. PLOS requires an ORCID iD for the corresponding author in Editorial Manager on papers submitted after December 6th, 2016. Please ensure that you have an ORCID iD and that it is validated in Editorial Manager. To do this, go to ‘Update my Information’ (in the upper left-hand corner of the main menu), and click on the Fetch/Validate link next to the ORCID field. This will take you to the ORCID site and allow you to create a new iD or authenticate a pre-existing iD in Editorial Manager. Please see the following video for instructions on linking an ORCID iD to your Editorial Manager account: https://www.youtube.com/watch?v=_xcclfuvtxQ.

Additional Editor Comments:

It needs Major Revision

Reviewers' comments:

Reviewer's Responses to Questions

**Comments to the Author**

1. Is the manuscript technically sound, and do the data support the conclusions?

Reviewer #1: Yes

Reviewer #2: Yes

2. Has the statistical analysis been performed appropriately and rigorously? 

Reviewer #1: Yes

Reviewer #2: Yes

3. Have the authors made all data underlying the findings in their manuscript fully available?

Reviewer #1: No

Reviewer #2: Yes

4. Is the manuscript presented in an intelligible fashion and written in standard English?

Reviewer #1: No

Reviewer #2: Yes

5. Review Comments to the Author

Reviewer #1: Dear authors thanks for this interesting review. In general, this review is well written. However, I have some comments and questions

1.Your review needs major language edition

2.Please incorporate your review Prospero registration number in abstract section

3.Please put the citation for the PRISMA guideline you used under method section. Is that PRISMA guideline 2020 or not?

4.If you used the PRSMA guideline 2020, some components were missed for example Certainty assessment

5.Please avoid the term data analysis and process in method section rather use Synthesis methods and Reporting bias assessment

6.Avoid a paragraph ‘’Fixed effect model would have been used 187 if studies were homogeneous’’

7.What were the limitations of your review?

8.Lastly, please use PRISMA 2020 https://pubmed.ncbi.nlm.nih.gov/33782057/ guideline and rewrite your review and meta-analysis

Thank you

Reviewer #2: Your manuscript is full of self-plagiarism from your previous published work (Kasahun AW, Zewdie A, Shitu S, Alemayehu G. Vaccine cold chain management practice and associated factors among health professionals in Ethiopia: systematic review and meta-analysis. Journal of Pharmaceutical Policy and Practice. 2023 Dec;16(1):1-0.).

You used a mnemonic PICO (population, intervention, control, outcome), wrongly. Since your review is on prevalence studies, you need to use CoCoPop (condition, context, population). I have doubt on your inclusion criteria since you use PICO. I need to see your search strategies and search terms that you use for each database.

On methods part line 118 needs citation.

You use Joanna Briggs Institute for data extraction, why not for quality assessment and analysis too?

6. PLOS authors have the option to publish the peer review history of their article (what does this mean?). If published, this will include your full peer review and any attached files.

Reviewer #1: **Yes: **Bekahegn Girma

Reviewer #2: No

---

## [Author Response · Author response to Decision Letter 0]

4 Jul 2023

Dear Academic editor of PLOS ONE and the reviewers, we are highly indebted to your important comments and queries of clarifications for enriching our manuscript. The following statements are tailored to address all of your reflections point by point in their order of appearance from the forwarded comments. 

1. Please ensure that your manuscript meets PLOS ONE's style requirements

A: we revised the manuscript in accordance with PLOS ONE style requirements 

2. In your Data Availability statement, you have not specified where the minimal data set underlying the results described in your manuscript can be found. PLOS defines a study's minimal data set as the underlying data used to reach the conclusions drawn in the manuscript and any additional data required to replicate the reported study findings in their entirety. All PLOS journals require that the minimal data set be made fully available. For more information about our data policy,

A: Our manuscript solely depends on the previous research works; as a result, we have not primary data set for it. But as additional information file, we will upload the extracted data set that we used for estimating the pooled prevalence of the outcome variable.

3. Please upload a new copy of Figure 2 and 4 as the detail is not clear.

A: we have uploaded clearer version of fig.2 and fig.4 

4. PLOS requires an ORCID iD for the corresponding author in Editorial Manager on papers submitted after December 6th, 2016. Please ensure that you have an ORCID iD and that it is validated in Editorial Manager. To do this, go to ‘Update my Information’ (in the upper left-hand corner of the main menu), and click on the Fetch/Validate link next to the ORCID field. This will take you to the ORCID site and allow you to create a new iD or authenticate a pre-existing iD in Editorial Manager. Please see the following video for instructions on linking an ORCID iD to your Editorial Manager account: 

A: ORCID iD of the corresponding author is updated and linked to to the Editorial Manager.

R1: Please incorporate your review Prospero registration number in abstract section

A; Thanks, we incorporated the PROSPERO registration number on the abstract section of the revised manuscript

R1: Please put the citation for the PRISMA guideline you used under method section. Is that PRISMA guideline 2020 or not?

A: Yes, we have used PRISMA 2020 guideline, and appropriate reference is cited on the revised version.

R1: If you used the PRSMA guideline 2020, some components were missed for example Certainty assessment

A: Thank you; we have included certainty assessment methods and its findings on the method and result section of the manuscript respectively.

R1: Please avoid the term data analysis and process in method section rather use Synthesis methods and Reporting bias assessment 

A: thanks, Corrected accordingly 

R1: Avoid a paragraph ‘’Fixed effect model would have been used 187 if studies were homogeneous’’

A: Thanks, removed on the revised version 

R1: What were the limitations of your review?

A: We included the limitations of our review at the end of the discussion on the revised version of the manuscript. The result of this systematic review and meta-analysis is based on few studies; as a result it may not show the true magnitude of the outcome variable. It may possibly different if many more studies were done and included in this meta-analysis and review. Above all, this result is solely depends on the already available data from previous studies, as a result the limitation and quality of the previous researches have an impact on our findings.

R1: Lastly, please use PRISMA 2020 https://pubmed.ncbi.nlm.nih.gov/33782057/ guideline and rewrite your review and meta-analysis

A: thank you, we have revised in accordance with the referred guideline (PRISMA2020)

R2: I have seen self-plagiarism from your previous published work (Kasahun AW, Zewdie A, Shitu S, Alemayehu G. Vaccine cold chain management practice and associated factors among health professionals in Ethiopia: systematic review and meta-analysis. Journal of Pharmaceutical Policy and Practice. 2023 Dec;16(1):1-0.).

A: Thanks, There was some text overlap from the previously published article on vaccine cold chain management practices as the condition understudy is closely related. We have made a meticulous revision on those issues on the revised version. 

R2: You used a mnemonic PICO (population, intervention, control, outcome), we use it for effectiveness studies to develop question, instead of using CoCoPop (condition, context, population) for prevalence studies. I have doubt on your inclusion criteria since you use PICO. I need to see your search strategies and search terms that you use for each database.

A: Thank you, we actually used CoCoPop/PEO, though it is wrongly mentioned as PICO in the submitted manuscript, unfortunately we retained the template that we have used for the other review. We have corrected it on the revised version of the manuscript. The objectives of our review are: to determine the pooled magnitude of health professionals’ knowledge on vaccine cold chain management and to identify its associated factors. As the review has prevalence and risk assessment objectives we have used CoCoPop/PEO guide.

Condition: Knowledge on vaccine cold chain management

Population: Health professionals

Context: Ethiopia

Exposure: Are socio-demographic, health facility and other factors that negatively or positively affect health professionals’ knowledge on vaccine cold chain management; it includes level of training, professional category, on job training, supervision and others.

Outcome: Health professionals’ knowledge on vaccine cold chain management which is determined using a composite of items

R2:. I need to see your search strategies and search terms that you use for each database.

A: Kindly find some of the search terms that we used in each data base. Furthermore, a hand search was made on references of retrieved studies using the mentioned search terms.

Pubmed

(((((((((((vaccine cold chain management) OR (vaccine cold chain[MeSH Terms])) OR (vaccine cold chain[Title/Abstract])) AND (knowledge[MeSH Terms])) OR (knowledge[Title/Abstract])) AND (health professionals[Title/Abstract])) OR (health professionals[MeSH Terms])) OR (vaccine handlers[MeSH Terms])) OR (vaccine handlers[Title/Abstract]))) AND (Ethiopia[Title/Abstract])) OR (Ethiopia[MeSH Terms])

Google scholar 

All in title: health professionals’ knowledge on vaccine cold chain management AND associated factors in Ethiopia.

All in title: "cold chain management" AND "health professionals" Ethiopia 

All in title: "health professionals’ knowledge on vaccine cold chain management” AND Ethiopia 

All in title: "vaccine cold chain management" AND Ethiopia

All in title: "vaccine cold chain" AND "health professionals knowledge" 

All in title: "vaccine handlers’ knowledge on vaccine cold chain management" AND Ethiopia

African Journals online

“Health professionals’ knowledge on vaccine cold chain management and associated factors in Ethiopia”

“Determinants of health professionals’ knowledge on vaccine cold chain management in Ethiopia”

Web of science

(((((((((((vaccine cold chain management) OR (vaccine cold chain[all fields])) OR (vaccine cold chain[all fields])) AND (knowledge[topic])) OR (knowledge[all fields])) AND (health professionals[all fields])) OR (health professionals[topic])) OR (vaccine handlers[all fields])) OR (vaccine handlers[topic]))) AND (Ethiopia[title])) OR (all fields])

R2: On methods part line 118 needs citation.

A: thank you, appropriate reference cited on the revised version

R2: You use Joanna Briggs Institute for data extraction, why not for quality assessment and analysis too?

A: We are familiar with the tools that we have used for assessing quality and synthesizing the data rather than JBI tools which we do not know much about. We have no other reason not to use JBI tools for quality assessment and analysis if any, we opt tools that we know better for managing and measuring variables.

---

## [Decision Letter · Decision Letter 1]

6 Oct 2023

Health professionals’ Knowledge on Vaccine cold chain management and associated factors in Ethiopia: systematic review and meta-analysis

PONE-D-23-03134R1

Dear Dr. Abebawe,

We’re pleased to inform you that your manuscript has been judged scientifically suitable for publication and will be formally accepted for publication once it meets all outstanding technical requirements.

Kind regards,

Demeke Mesfin Belay, 

Academic Editor

PLOS ONE

Additional Editor Comments (optional):

Congratulations on your acceptance with minor revisions!

Please take into account the following constructive feedback to enhance the quality of your manuscript.

It would be beneficial to include the quality assessment result as a supplementary file. Additionally, please address the potential reasons for publication bias and provide the findings of the subgroup analysis in your discussion.

Reviewers' comments:

Reviewer's Responses to Questions

**Comments to the Author**

1. If the authors have adequately addressed your comments raised in a previous round of review and you feel that this manuscript is now acceptable for publication, you may indicate that here to bypass the “Comments to the Author” section, enter your conflict of interest statement in the “Confidential to Editor” section, and submit your "Accept" recommendation.

Reviewer #2: All comments have been addressed

2. Is the manuscript technically sound, and do the data support the conclusions?

Reviewer #2: Yes

3. Has the statistical analysis been performed appropriately and rigorously? 

Reviewer #2: Yes

4. Have the authors made all data underlying the findings in their manuscript fully available?

Reviewer #2: Yes

5. Is the manuscript presented in an intelligible fashion and written in standard English?

Reviewer #2: Yes

6. Review Comments to the Author

Reviewer #2: (No Response)

7. PLOS authors have the option to publish the peer review history of their article (what does this mean?). If published, this will include your full peer review and any attached files.

Reviewer #2: No

---

## [Editor Report · Acceptance letter]

14 Nov 2023

PONE-D-23-03134R1 

Health professionals’ Knowledge on Vaccine cold chain management and associated factors in Ethiopia: systematic review and meta-analysis 

Dear Dr. Kasahun:

I'm pleased to inform you that your manuscript has been deemed suitable for publication in PLOS ONE. Congratulations! Your manuscript is now with our production department. 

Kind regards, 

on behalf of

Mr. Demeke Mesfin Belay 

Academic Editor

PLOS ONE